# Does Social Capital Benefit the Improvement of Rural Households' Sustainable Livelihood Ability? Based on the Survey Data of Jiangxi Province, China

Feixue Xiong [1], Shubin Zhu [2,*], Hui Xiao [3], Xiaolan Kang [2] and Fangting Xie [2,*]

[1] School of Economics and Management, Jiangxi Agricultural University, Nanchang 330045, China; feixuexiong@163.com
[2] Institute of New Rural Development, Jiangxi Agricultural University, Nanchang 330045, China; kxl510@163.com
[3] School of Economics & Management, Beijing Forestry University, Beijing 100083, China; huixiao9608@163.com
* Correspondence: shubinzhu@163.com (S.Z.); fountain_xie@163.com (F.X.)

**Abstract:** This article examines the influence of social capital on the sustainable livelihood ability of rural households who are out of poverty, in order to promote the sustainable development of their livelihood. Based on the survey data of 371 out-of-poverty households in rural Jiangxi, we analyzed the relationship between social capital and households' sustainable livelihood ability using "Ordinary Least Square (OLS) + robust standard error" regression models and quantile regression models. Households' social capital was measured from the following three dimensions: social network, social participation, and social trust. The benchmark regression models showed that social capital index, social network, and social participation all had a significant positive effect on the sustainable livelihood ability of out-of-poverty households. However, the impact of social trust on sustainable livelihood ability was not significant. In addition, the quantile regression analysis results showed that social capital index, social network, social participation, and social trust all contributed the most to households with a low sustainable livelihood ability. Therefore, it is suggested to improve the social capital accumulation of out-of-poverty households from multiple dimensions, so as to enhance the sustainable livelihood ability of households and consolidate poverty-alleviation achievements.

**Keywords:** social capital; sustainable livelihood; out-of-poverty households

## 1. Introduction

Poverty exists in every country, which hinders the development of human civilization [1,2]. In fact, it can be said that the history of human development is actually an anti-poverty process [3]. Since the reform and opening up more than 40 years ago, China hasissued and implemented a large number of pertinent policies to anti-poverty and poverty alleviation. Among them, the *Decision on Winning the Fight against Poverty* issued by the Central Committee of the Communist Party of China and the State Council in 2015 clearly stated that more than 70 million rural poor people should live out of poverty by 2020. In 2020, the country had made "unprecedented achievements" in fighting poverty, eliminating absolute poverty, and solving regional overall poverty. However, the economic development foundation of poverty-stricken areas is still weak, the employment of out-of-poverty laborers is still unstable, and some of the out-of-poverty households still have the risk of returning to poverty. This indicates that poverty alleviation should change from eliminating absolute poverty to alleviating unbalanced and insufficient relative poverty [4]. Poverty alleviation is not only a temporary result, but a sustainable state. How to get rid of poverty stably and sustainably is a new problem in society. In the critical period of effective connection between poverty alleviation and rural revitalization, the sustainable livelihood development of out-of-poverty households is the primary task in the work of the

"Three Issues Concerning Farmers". Therefore, it is of important practical value to study the sustainable livelihood of the out-of-poverty households.

Social capital, which was first formally proposed by Hanifan (1916) [5], refers to goodwill, friendship, compassion, and social interaction between a group of individuals and families. Since then, many studies on social capital have entered the scholars' vision. Bourdieu [6] (from the perspective of relational network), Coleman [7] (from the perspective of social structure), Putnam [8] (from the perspective of social organization), Portes [9] (from the perspective of structural holes) and other scholars have explained it from different perspectives. Since 1978, China has formed a relatively complete multi-coordinated poverty alleviation mechanism among government, market, and society. Under this mechanism, social capital is the decisive factor for poverty alleviation in poor areas [10]. Traditional poverty alleviation is mainly carried out by investing a large amount of capital and material resources, but it is still ineffective in some areas. Grootaert (1998) [11] noted that the poor effectiveness of financial poverty alleviation was due to the lack of social capital linking material input. In 2000, the World Bank claimed that social capital was an important part of the anti-poverty process and the third largest capital after physical capital and human capital. In fact, poverty refers to the lack of development capacity and low income level. Additionally, it refers to the lack of social status, social ties, and social opportunities in social, economic, and political activities. Gradually, it may result in households' marginalization. The lack of necessary social capital will restrict households' access to necessary information and opportunities, limit their social mobility, and affect employment and income. A deficiency of social support will lead to subjective poverty and objective poverty [12], which may result in unstable out-of-poverty households returning to poverty. It can be seen that social capital has practical significance for poverty alleviation and a sustainable livelihood.

At present, research on the impact of social capital on sustainable livelihoods mainly focuses on the following aspects: poverty alleviation [13], income [14], sustainable livelihoods [15], livelihood vulnerability [16], etc. Scholars have measured social capital and sustainable livelihoods from different perspectives according to their respective concerns (Table 1). Most of the studies believe that social capital can reduce poverty vulnerability, alleviate poverty, prevent the intergenerational transmission of poverty, and increase income, thus improving the sustainable livelihoods of households. The main ways that social capital affects the sustainable livelihood ability of households are as follows: promoting the labor mobility of rural households [17], easing financing constraints [18], improving body quality and health [19], increasing farmland rent rate and agricultural production efficiency [20,21], and expanding the availability of entrepreneurial resources [22], etc.

**Table 1.** Indicator selection of related social capital research.

| Research Perspective | Dependent Variable | Independent Variable (Social Capital) | Study Region | Research Object | Researchers |
|---|---|---|---|---|---|
| Social capital and poverty reduction | Poverty | Political ties, business ties, associational membership, institutional trust | Western China | Households of neighborhood committee or township | Zhang et al. [13] |
| | Multidimensional poverty | Number of memberships in groups, number of active members in formal groups, number of active members in informal groups | Rural Vietnam | Rural households | Pham et al. [23] |
| | Intergenerational transmission of poverty | Whether or not any family member holds the post of state functionary or village cadre, whether or not families receive assistance when they are in trouble | Six poverty-stricken counties of China | Rural households | Wu et al. [24] |
| Social capital and income | Income | Village social capital, family social capital | China | Rural households | Zhou [14] |
| | Income inequality | Robert Putnam's state-level social capital index (SCI) | The United States | | Liu et al. [25] |

**Table 1.** *Cont.*

| Research Perspective | Dependent Variable | Independent Variable (Social Capital) | Study Region | Research Object | Researchers |
|---|---|---|---|---|---|
| Social capital and sustainable livelihoods | Sustainable livelihood index | Exposure, sensitivity, adaptive capacity | Koshi River basin community in Nepal | Rural households | Zhang et al. [15] |
| Social capital and livelihood vulnerability | Livelihood shocks | Group memberships, reciprocity, social capital index | Rural South Africa | Rural households | Mbiba et al. [16] |
| | Livelihood resilience index | Cooperation of family, influence or political power in the village, participation in non-governmental, participate in agriculture or tree planting group, communication with neighbors | Bakhtegan basin, Iran | Rural house-holds | Nasrnia et al. [26] |
| | Livelihood Sensitivity | Number of organization membership, number of participation, activeness score, bonding social capital based organizations, linking social capital based organizations, bridging social capital | Northeastern Floodplains of Bangladesh | Rural house-holds | Tuihedur et al. [27] |
| | Livelihood vulnerability | Community cohesion and networks, gender equity, decision making, leadership | Rural coastal communities of Solomon Islands | Rural house-holds | Malherbe et al. [28] |
| | Livelihood adaptation ability | Number of telephone contacts, number of relatives, village cadres, time to the furthest household in a village | Loess Plateau, China | Rural house-holds | Li et al. [29] |

However, can social capital benefit each individual in the society equally? Although most scholars agree with the positive role of social capital, its impact on the poor is still controversial. On the one hand, some scholars believed that social capital helped to reduce the risk of household livelihood instability and prolong the time of livelihood stability in poor areas [30]. On the other hand, some scholars pointed out that social capital did not belong to the poor, and it did not have a significant effect on the improvement of households' poverty reduction and sustainable livelihood ability [14]. Social capital had an obvious threshold effect on household income. Only when social capital was above the threshold value, could it significantly increase the income of households [31]. Yang et al. found that rural households faced the least social pressure and had the strongest adaptability in social capital. However, social pressure and social capital did not significantly affect their choice of livelihood strategy [32]. Moreover, social capital would exacerbate inequality [33]. Due to different factors, such as natural disasters, agricultural development level, and topography, the vulnerability of households' social capital in different regions was differently unequal [34]. When the social capital of rural households could not completely alleviate the negative shock on livelihoods, possibly because social capital needed to work with physical capital to produce positive synergies [16]. In addition, the resource acquisition role of social capital depended on individual power and status [6]. Social capital and inequality were considered interrelated [30]. As a result of unequal power and status, social capital might continue to consolidate the disadvantage of households, and, thus, poverty [34].

At present, the influence of social capital on sustainable livelihood ability remains unclear. In particular, there are still few studies on the special group of out-of-poverty households (Table 1). In summary, can social capital promote the improvement of the sustainable livelihood ability of poverty-stricken households? On which of the different sustainable livelihood ability levels does social capital play the greatest positive role? Considering this, in this study, we used the survey data of out-of-poverty households in Jiangxi Province, and applied the "OLS + robust standard error" regression model and quantile regression model to explore the impact of social capital on sustainable livelihoods. Furthermore, we explored the impact of social capital on sustainable livelihoods at different

levels from the perspectives of social networks, social participation, and social trust in order to provide a valuable reference for consolidating and expanding the achievements of poverty alleviation.

The structure of this paper is as follows. The first part is the introduction, explaining the background of poverty alleviation in China and sorting out the pertinent literature on social capital and sustainable livelihood ability. The second part expounds the theoretical mechanism of social capital's impact on sustainable livelihood ability and puts forward research hypotheses. The third part introduces the data, variables, and empirical methods used in this paper. The fourth part reports the empirical results of the impact of social capital and its three dimensions on the sustainable livelihood of out-of-poverty households, and carries out the robustness test. The fifth part summarizes the research conclusions and discusses the corresponding policy implications.

## 2. Analytical Framework

The term "livelihoods" began with the study of poverty, which refers to the way or means of making a living. The internationally recognized concept of livelihood includes core elements such as active, assets, and capacity [35], with more emphasis on the sustainability livelihood. With the deepening of the connotation and scope of livelihood research, scholars put forward the framework of sustainable livelihood analysis. Among them, the most representative and most widely used is the sustainable livelihoods analysis framework of the Department for International Development (DFID) [36]. The DFID framework was outlined in *Sustainable Livelihoods Guidance Sheets*. It is a systematic approach to the development of sustainable livelihoods for poor households, highlighting key factors influencing poverty. The core content of the sustainable livelihood framework is livelihood capital. It contains human capital, financial capital, social capital, material capital, and natural capital [37]. Poor households can use certain capital or a variety of capital combinations to optimize their livelihood strategies and achieve a sustainable livelihood. Based on the DFID framework, this paper studies the sustainable livelihood development of out-of-poverty households with social capital as the core.

Livelihood capital is the foundation and the key to the sustainable development ability of poor households. According to the theory of asset poverty reduction, capital is defined as the stock of financial, human, natural, and social resources that can be acquired, developed, and improved. They are not only the resources people use to make a living, but also give people the ability to plan and act [38]. As a stock, livelihood capital can generate income flow, consumption, and additional storage. Rural China is a typical human society. As an important livelihood capital of households, social capital is often regarded as an informal insurance system, which has an important impact on the sustainable development of households.

With the deepening of research, scholars generally realized that the core element of social capital was social network, social participation, and social trust [39,40]. Therefore, starting from the three dimensions of social network, social participation, and social trust, we constructed the analytical framework of social capital and sustainable livelihood ability (Figure 1).

As one of the core dimensions of social capital, social network is composed of embedded relationships among members of the common organization. As an interpersonal resource network, it has relative stability. The social network of households is a network relationship with kinship as the main axis, which is a kind of differential pattern [41]. Social networks can effectively solve the problem of information asymmetry. By transferring high-quality and reliable information between friends and acquaintances, they can increase employment opportunities, share livelihood risks, increase income, and achieve sustainable development [13]. According to a study by Shao et al. [42], the social network had a significant positive impact on farmers' income, and formal financial loans played a positive mediation effect.

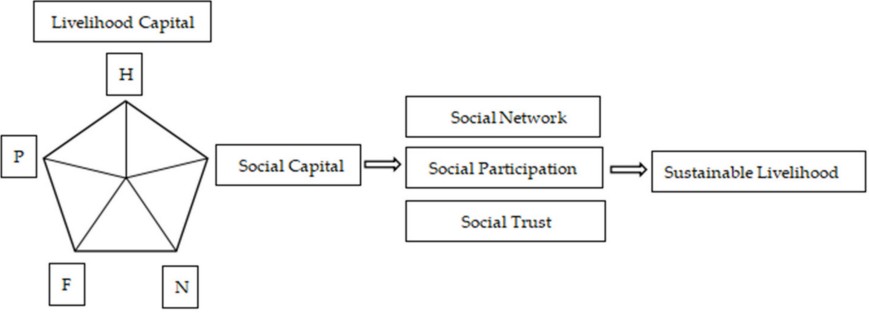

**Figure 1.** Chart of sustainable livelihood analytical framework for social capital.

Social participation refers to the participation of social members in socio-economic organizations. The more actively households participate in various group activities in the village, the more opportunities and resources they will have, and the stronger their ability to avoid risk impact [17]. The strong guarantee of social participation has a great mobilization effect on households' production enthusiasm and interest perception. By improving households' self-efficacy, it can optimize their own production behavior, adjust their livelihood strategies, and improve their income and sustainable livelihood ability [43,44].

Social trust is defined as a trust relationship that can perceive the expected behavior of others, which usually develops between interacting individuals. Rural China is a typical acquaintance society with a high degree of trust between households. As the key to strengthening social communication, social trust can reduce transaction costs and information asymmetry and risks, thereby improving efficiency and productivity [45]. Trust-based social relations can effectively avoid dishonest behavior and free rider problems [46,47]. Individual trust in organizations and institutions reflects the quality of local governance [48]. To sum up, social capital mainly responds to livelihood shocks and maintains a sustainable livelihood through these three dimensions [49–51].

In summary, we propose the following hypotheses:

**Hypothesis 1.** *Social capital has a significant positive impact on the sustainable livelihood ability of households out of poverty.*

**Hypothesis 2.** *Social network has a significant positive impact on the sustainable livelihood ability of households out of poverty.*

**Hypothesis 3.** *Social participation has a significant positive impact on the sustainable livelihood ability of households out of poverty.*

**Hypothesis 4.** *Social trust has a significant positive impact on the sustainable livelihood ability of households out of poverty.*

## 3. Materials and Methods

### 3.1. Research Area and Sample Selection

As the former Central Soviet Area and the old revolutionary base area, Jiangxi Province has a special status and distinctive characteristics in the process of poverty alleviation due to geographical and historical reasons. It is the main battlefield for poverty alleviation in China. There are 54 former Central Soviet counties, 17 Luoxiao Mountain counties, and 25 poverty-stricken counties in the province. Among them, there are 2900 poor villages, of which 269 are extremely poor villages [52]. Overall, the economic development of Jiangxi Province is relatively backward, the basic conditions are weak, and the overall regional poverty is prominent. In 2020, Jiangxi Province won the battle against poverty with high quality, as scheduled. Additionally, 801,000 poverty-stricken households with 2.816 million people were lifted out of poverty. The per capita income of out-of-poverty households

increased from 3344 yuan in 2015 to 12,626 yuan in 2020. The regional poverty in the whole province has been solved historically [53].

We chose Nanchang City of Jiangxi Province as the investigation area for the following three reasons. Firstly, the sustainable livelihood of out-of-poverty households in this area is special. This region belongs to the provincial capital city of Jiangxi Province, and the progress of households' poverty alleviation has been in the forefront of the province. Secondly, the sustainable livelihood of the households in this region is typical. The assistance provided by the government to the poor households is very typical. Additionally, each poor household has docking and helping cadres. Thirdly, the survey area covers 5 counties, covering plains, hills, and mountainous areas. The geographical scope is wide, and the landform types are rich and diverse. In addition, poverty varies greatly among counties. Therefore, the survey samples in the region are universal, and can fully reflect the situation of different landform types and different levels of poverty.

In 2015, there were 36,517 people who lived in poverty in Nanchang, which decreased to 1822 in 2019. The poverty ratio dropped from 2.00% in 2015 to 0.07% in 2019. By June 2020, all the poor households and the poor people in the city were out of poverty. The per capita disposable income of the poor increased from 4102 yuan in 2015 to 10,459 yuan in 2019. The proportion of the per capita disposable income of the poor to that of the rural households in Nanchang increased from 29.96% in 2015 to 53.64% in 2019.

The data used in this study come from a sample survey of households in Nanchang of Jiangxi Province from June to December 2020. In this research, poor households refer to those with poverty files and cards established by the government. In 2013, households whose annual net income per capita was less than 2736 yuan were identified as poor households. In 2020, the poverty standard was increased to 4000 yuan. Out-of-poverty households refer to those families with an annual net income per capita that exceeds the poverty recognition standard. Additionally, they are free from worry about food and clothing, and have access to compulsory education, basic medical services, and safe housing. The survey used stratified sampling and random sampling to select survey objects based on the geographic location, economic development level, and number of poor households in each township. The survey area involved 49 administrative villages (including 30 poor villages and 19 non-poor villages), 20 townships (towns), and 5 counties (districts). The data collected involved the following three levels: individual characteristics, family characteristics, and village characteristics. The investigators of the research adopted household interviews to conduct a comprehensive survey of the status of family members, family capital status, livelihood activities, and other aspects of the out-of-poverty households. A total of 371 valid questionnaires were collected. During the survey, all sample households had been lifted out of poverty, a total of 1058 people.

*3.2. Selection and Definition of the Model Variable*

The purpose of this study was to analyze the influence of social capital and its different dimensions on the sustainable livelihood ability of out-of-poverty households. In the setting of a dependent variable, this study selected "the proportion of income other than transfer income (such as pensions and subsidies granted by the government or charitable organization) to the total net income of the household in 2019" to measure the sustainable livelihood ability of out-of-poverty households. Income other than transfer income can directly reflect the sustainable livelihood ability. The income structure of households can be divided into operational income, wage income, transfer income, and property income. Among them, the transfer income includes five guarantees, minimum guarantees, government assistance, subsidies for the disabled, etc. It is manifested as the "blood transfusion" role of the government in poverty alleviation. Income other than transfer income is the real expression of endogenous motivation of the out-of-poverty households, and has the "blood-making" effect of sustainable livelihood. Except for transfer income, the proportion of other income can reflect the sustainable livelihood ability of them well.

In the selection of independent variables, the social capital of out-of-poverty households is the focus of this research. In quantitative analysis, social capital is a difficult variable to observe and measure. This article refers to the classification of Putnam (1994) [8] and divides social capital into the following three dimensions: social network, social participation, and social trust.

As for social network, due to the traditional agricultural culture, an acquaintance society based on kinship, clan, and geographic relationships has been formed in China's rural areas. The wider social network of out-of-poverty households is, the more non-agricultural employment opportunities they will obtain. Non-agricultural employment is an important means for out-of-poverty households to obtain a higher income and prevent poverty. It can improve their ability to resist risks and sustainable livelihoods. Based on this, we selected "the number of families that can provide help when out-of-poverty households looking for non-agricultural jobs (*S-network*)" to measure the social network of households.

In terms of social participation, as the main carrier of industrial poverty alleviation, farmers' professional cooperatives have a significant pro-poor nature. Out-of-poverty households participating in farmers' professional cooperatives can not only solve work problems, but also receive policy dividends. Based on this, this paper selected "participation in planting associations, cooperatives, and other organizations (*S-part*)" to measure the social participation of households.

Social trust is the foundation of social network, and social network is the condition of social trust. Familiar villagers have localized emotional identification and trust. They communicate closely and the information between them is transparent. This kind of social trust can reduce transaction costs and make it easy to reach a cooperation contract. The interpersonal assistance of high-frequency interaction villagers can reduce the livelihood risks of poverty-stricken households. Therefore, we selected "trust in local villagers (*S-trust*)" to measure the social trust of households.

We considered that there might be an endogenous problem in the relationship between households' social capital and income. We referred to Zhou's research (2012) [14] and used the entropy method to weigh the dimensional variables of social capital into a comprehensive social capital index to reduce the impact of endogeneity. The specific calculation formula is as follows:

$$S - index_i = \sum_{k=1}^{n} w_k S - k \tag{1}$$

Among them, $w_k$ stands for the weight of each dimension of social capital (Table 2), $S - k$ (k = 1, 2, 3) is the data of each social capital variable after the normalization of the minimax method.

**Table 2.** The weights of social capital variables.

| Variable | S-network | S-part | S-trust |
|:--------:|:---------:|:------:|:-------:|
| **Weight** | 0.421 | 0.460 | 0.119 |

In the framework of sustainable livelihoods, livelihood capital factors that affected the sustainable livelihood ability also included human capital, natural capital, physical capital, and financial capital. In addition, this paper also selected characteristic variables at the village level in order to control the error of estimation result caused by omission of related variables. Referring to the existing literature [15,26,54], this study selected the following control variables. Human capital characteristics included family human capital (the number of family laborers) and household head human capital (years of education of the household head, whether the household head obtained agricultural or non-agricultural training). Natural capital was indicated by the area of cultivated farmland. Physical capital was represented by "building area of house". Additionally, financial capital was measured

by "whether the household obtained loan from bank". Regional environmental factors mainly included "the number of poverty-stricken people in the local village". In Table 3, we report the definitions, values, and descriptive statistics for all variables.

**Table 3.** Definition of variables and descriptive statistics.

| Variable | Definition | Max | Min | Mean | SD |
|---|---|---|---|---|---|
| | **Dependent variable** | | | | |
| *SLC* | The proportion of income other than transfer income to the total net income of the household in 2019 (%) | 97.66 | 1.20 | 52.74 | 24.83 |
| | **Independent variables** | | | | |
| *S-index* | The weighted value of three dimensional variables of social capital (Equation (1)) | 1 | 0 | 0.33 | 0.24 |
| *S-network* | The number of families that can provide help when out-of-poverty households are looking for non-agricultural jobs (households) | 30 | 0 | 3.08 | 4.06 |
| *S-part* | Participation in planting associations, cooperatives and other organizations (1 = yes, 0 = no) | 1 | 0 | 0.49 | 0.50 |
| *S-trust* | Trust in local villagers (1 = great distrust, 2 = distrust, 3 = general, 4 = trust, 5 = great trust) | 5 | 1 | 2.96 | 1.09 |
| *Labor* | The number of family laborers (person) | 5 | 0 | 0.92 | 1.06 |
| *Education* | Years of education of the household head (years) | 16 | 0 | 4.24 | 3.38 |
| *Train* | Whether the household head obtained agricultural or non-agricultural training (1 = yes, 0 = no) | 1 | 0 | 0.13 | 0.34 |
| *A-land* | The area of cultivated farmland (mu) | 132 | 0 | 4.24 | 8.14 |
| *A-housing* | Building area of house (square meters) | 600 | 15 | 121.90 | 78.33 |
| *Loan* | Whether the household obtained a loan from a bank (1 = yes, 0 = no) | 1 | 0 | 0.20 | 0.40 |
| *V-poor* | The number of poverty-stricken people in the local village (person) | 100 | 5 | 49.09 | 22.25 |

*3.3. Model Setting*

We analyzed the impact of social capital on sustainable livelihood ability of out-of-poverty households in two steps. In the first step, the "Ordinary Least Squares + robust standard error" benchmark model was used to examine the impact of social capital and each dimension on the sustainable livelihood ability of households. The "OLS" method was based on the mean values of the variables. It used the function of the conditional mean values of the dependent variables to describe the mean values of the dependent variables under each specific value of the independent variables. Accordingly, the relationships between the independent variables and the dependent variables were revealed. The measurement model is as follows:

$$SLC_i = \beta_0 + \beta_s S_i + \beta_h H_i + \beta_n N_i + \beta_p P_i + \beta_f F_i + \beta_v V_i + \varepsilon_i \tag{2}$$

Among them, $SLC_i$ represents the sustainable livelihood ability of out-of-poverty household $i$. $S_i$ is the core variable of this paper, and it represents the social capital of the household $i$, including social capital composite index, social network, social participation, and social trust. $H_i$, $N_i$, $P_i$, $F_i$, and $V_i$ are control variables denoting human capital, natural capital, physical capital, financial capital, and village characteristics, respectively. $\varepsilon_i$ represents random interference.

In the second step, based on the analysis of the benchmark model, we used quantile regression to analyze the effects of social capital index, social network, social participation, and social trust on sustainable livelihood ability at different levels. The quantile regression was proposed by Koenker and Bassett (1978) [55]. It was used to estimate the linear relationships between a set of regression variables and the quantiles of explained variables. Additionally, it emphasizes the change of conditional quantile. The least square estimation assumes that the explanatory variable can only affect the mean position of the conditional distribution of the explained variable. However, the quantile regression estimation can

accurately describe the influence of the explanatory variables on the variation range and the conditional distribution shapes of the explained variables. It is denoted that the overall $q$ quantile $y_q(x)$ of the conditional distribution $y \mid x$ is a linear function of $x$, as follows:

$$y_q(x_i) = x_i' \beta_q \tag{3}$$

Among them, $y$ is sustainable livelihood ability of out-of-poverty households. $x_i$ is the vector of independent variables, including core independent variables and control variables. $\beta_q$ is the "$q$ quantile regression coefficient", whose estimator $\beta_q^{\wedge}$ can be defined by the following minimization problem:

$$\min_{\beta_q} \sum_{i:y_i \geq x_i' \beta_q}^{n} q \left| y_i - x_i' \beta_q \right| + \sum_{i:y_i < x_i' \beta_q}^{n} (1-q) \left| y_i - x_i' \beta_q \right| \tag{4}$$

This paper selected five representative quantiles of 0.1, 0.5, and 0.9, and used Stata 16.0 to iterative 500 times with the bootstrap method for quantile regression.

## 4. Empirical Results and Analysis

### 4.1. Descriptive Statistical Analysis

Table 3 shows data characteristics of sample out-of-poverty households. The details are as follows: the maximum sustainable livelihood ability of households is 97.66%, and the minimum is 1.2%. There is a big difference between households. Average sustainable livelihood ability is 52.74%, indicating that most out-of-poverty households still live on transfer income.

As for social capital, the average value of the social capital index is 0.33, and the overall level is not high. Social capital stock of households is quite heterogeneous, which shows that the maximum value of the social capital index is 1, and the minimum value is 0. From the perspective of the three dimensions of social capital, the average value of social network is 3.08. When farmers want to find non-agricultural jobs, not many people can help. It shows that the social network resources of poverty alleviation households are not rich. The average value of social participation is 0.49, implying that 49% of households have participated in economic organizations such as farmers' cooperatives. The trust level of the households in the local villagers is medium, with an average value of 2.96.

Regarding human capital characteristics, the number of laborers per household is 0.92, and the labor factor is relatively lacking. The average length of education of the household heads is 4.24 years. Among them, 20.49% of them have not received education, 44.74% of them have received elementary education, 21.83% of them have received junior high school education, and 12.94% of them have received high school education or above. The average value of whether the household heads have received agricultural or non-agricultural training is 0.13. That is, 13% of the household heads have received training. With respect to natural capital, the average cultivated area of farmland per household is 4.24 mu. With respect to physical capital, the average building area per household is 121.90 square meters. With respect to financial capital, about 20% of the out-of-poverty households have conducted regular financial lending. With respect to the village environment, the average number of poverty-stricken people in the local village is 49.09.

### 4.2. Benchmark Estimation Results

Table 4 showed the basic model results of the impact of social capital on the sustainable livelihood ability of out-of-poverty households. Models one, two, three, and four are econometric models with social capital composite index, social network, social participation, and social trust as the core independent variables, respectively. Based on the benchmark estimation results, this paper analyzed the impact of social capital on sustainable livelihood ability from the following three aspects: social capital index, the three dimensions of social capital, and control variables.

**Table 4.** Benchmark regression based on "OLS + robust standard error".

| Variable | Model 1 | Model 2 | Model 3 | Model 4 |
|---|---|---|---|---|
| S-index | 19.509 *** (4.812) | - | - | - |
| S-network | - | 0.759 *** (0.219) | - | - |
| S-part | - | - | 7.617 *** (2.340) | - |
| S-trust | - | - | - | 1.226 (1.014) |
| Labor | 9.363 *** (1.177) | 9.393 *** (1.196) | 9.569 *** (1.172) | 9.655 *** (1.201) |
| Education | 0.190 (0.321) | 0.274 (0.323) | 0.260 (0.323) | 0.297 (0.333) |
| Train | 5.228 (3.409) | 5.283 (3.371) | 6.161 * (3.384) | 6.681 ** (3.351) |
| A-land | 0.220 *** (0.077) | 0.281 *** (0.071) | 0.227 *** (0.078) | 0.277 *** (0.076) |
| A-housing | 0.044 *** (0.017) | 0.039 ** (0.017) | 0.045 *** (0.016) | 0.040 ** (0.017) |
| Loan | 4.582 * (2.715) | 6.143 ** (2.657) | 5.577 ** (2.728) | 6.644 ** (2.733) |
| V-poor | −0.074 (0.049) | −0.090 * (0.050) | −0.077 (0.050) | −0.091 * (0.051) |
| Constant | 32.624 *** (3.625) | 36.980 *** (3.471) | 34.413 *** (3.569) | 35.109 *** (4.160) |
| R-squared | 0.3058 | 0.2875 | 0.2957 | 0.2764 |
| Sample size | | | 371 | |

Note. Robust standard errors are in parentheses. * Significant at $\alpha = 0.10$, ** Significant at $\alpha = 0.05$, *** Significant at $\alpha = 0.01$.

In model one, the impact of social capital index on sustainable livelihood ability was significant at the 1% significance level. Additionally, its regression coefficient was 19.509. This indicates that the social capital index is conducive to improving the sustainable livelihood ability of households. This is consistent with the conclusions of most scholars [56,57]. Therefore, hypothesis one is validated.

With respect to the impact of the three dimensions of social capital on sustainable livelihood ability, both social network and social participation had a significant positive impact on sustainable livelihood ability, but social trust had no significant impact on sustainable livelihood ability. In model two and model three, the coefficients of social network and social participation were found to be significant at the 1% level or better. The regression coefficients were 0.759 and 7.617, respectively. In summary, hypothesis two and hypothesis three can be accepted, and hypothesis four is rejected.

From the perspective of human capital characteristics, *Labor* and *Train* were found to be positively associated with sustainable livelihood ability, while *Education* had no significant effect on sustainable livelihood ability (models one, two, three, and four). Through the allocation of family labor resources, farmers can enter the employment field with higher labor productivity and invest in high-quality agricultural production. The supply of labor fully guarantees the sustainable livelihood of out-of-poverty households. The training has improved the technical ability of farmers and expanded their career choices. It provides them with greater employ ability, more job opportunities, and more stable employment.

From the perspective of the natural capital characteristic, *A-land* was significantly positively correlated with sustainable livelihood ability (models one, two, three, and four). Cultivated land, as the most important natural capital of households, is the direct income source for out-of-poverty households. Generally, out-of-poverty households rely on natural capital to a high degree.

As for the physical capital characteristic, *A-housing* was significantly positively cor-related with sustainable livelihood ability (models one, two, three, and four). The size of the building area of a house can reflect the economic ability of out-of-poverty households. As the main loan collateral, the larger the building area of the house is, the stronger the repayment ability of them is, and the greater the probability of obtaining financial loans is.

Furthermore, from the perspective of financial capital, *Loan* was significantly positively correlated with sustainable livelihood ability (models one, two, three, and four). Financial capital can provide economic opportunities for households. When they encounter risk shock, capital borrowing can improve their ability to cope with risks, thereby enhancing their sustainable livelihood ability.

From the perspective of the village environmental factor, *V-poor* was found to be negatively associated with the sustainable livelihood ability of households (models two and four). The higher the number of poor people in villages is, the lower the economic development level of the village is, and the lower the sustainable livelihood ability of households is.

### 4.3. Quantile Regression of the Impact of Social Capital on Sustainable Livelihood Ability of Out-of-poverty Households

In order to fully consider the impact of social capital on sustainable livelihood ability at different levels, we further used the quantile regression model to investigate the impacts on the basis of models one, two, three, and four (Table 5). We selected three representative points, namely, 0.10, 0.50, and 0.90 points. They represented different levels of sustainable livelihood ability from low to high, namely the low sustainable livelihood ability group, the medium sustainable livelihood ability group, and the high sustainable livelihood ability group. The estimated coefficients of the related control variables were not significantly different from Table 4 in size and direction. Therefore, we omitted the coefficients of the control variables for simplicity in Table 5. As shown in Table 5, the impacts of social capital index, social network, social participation, and social trust on the sustainable livelihood ability of different levels were the same as that of the benchmark model. It was just that there was a difference between the size of the effect and the significance.

**Table 5.** Social capital quantile regression.

| Model | Variable | P = 0.10 | P = 0.50 | P = 0.90 |
|-------|----------|----------|----------|----------|
| Model 5 | *S-index* | 24.082 *** (8.095) | 14.257 ** (6.640) | 12.127 (8.009) |
| | Pseudo R$^2$ | 0.1366 | 0.2158 | 0.0998 |
| Model 6 | *S-network* | 1.225 * (0.739) | 0.880 *** (0.261) | 0.474 * (0.257) |
| | Pseudo R$^2$ | 0.1239 | 0.2198 | 0.1052 |
| Model 7 | *S-part* | 11.347 *** (4.057) | 6.073 * (3.342) | 4.170 (3.648) |
| | Pseudo R$^2$ | 0.1306 | 0.2100 | 0.0913 |
| Model 8 | *S-trust* | 3.653 * (1.982) | 1.279 (1.477) | 1.863 (1.665) |
| | Pseudo R$^2$ | 0.1185 | 0.2045 | 0.0919 |
| Control variables | | yes | yes | yes |
| Sample size | | | 371 | |

Note. Robust standard errors are in parentheses. The "yes" indicates that the variable is controlled. * Significant at $\alpha$ = 0.10, ** Significant at $\alpha$ = 0.05, *** Significant at $\alpha$ = 0.01.

From the quantile regression results of model five, we could see that as the quantiles rose, the quantile regression coefficients of the social capital index dropped from 24.082 to 14.257 and then rose to 19.044. At the 0.1 and 0.5 quantiles, the estimated coefficients passed the test at the significance level of 1 and 5%, respectively. However, the estimated coefficient failed the significance test at the 0.9 quantile. This shows that the social capital index has no significant impact on high sustainable livelihood ability and has a small impact on medium

sustainable livelihood ability. The biggest beneficiaries are out-of-poverty households with a low sustainable livelihood ability.

From the regression results of model six, social network had a significant positive effect on sustainable livelihood ability at the 0.1, 0.5, and 0.9 points. From the perspective of the size of the estimated coefficients, at the 0.1 quantile, the estimated coefficient was the largest (1.225), followed by the 0.5 quantile (0.880), and finally the 0.9 quantile (0.474), showing a downward trend. This shows that social network has the greatest effect on households with a low sustainable livelihood ability.

In model seven, social participation passed the 1% significance test at the 0.1 quantile and 10% at the 0.5 quantile. As the quantiles increased (0.1→0.5→0.9), the social participation coefficients showed a gradual decline (11.347→6.073→4.170). In other words, increasing social participation has the greatest impact on households with a low sustainable livelihood ability.

In model eight, social trust only passed the significance test at the 0.1 quantile. As the quantiles increased, the regression coefficients of the social trust showed a trend of falling first and rising (3.653→1.279→1.863).

*4.4. Robustness Test*

Aiming at the empirical results of the sustainable livelihoods of out-of-poverty households, this paper used the following three methods to test the robustness. Firstly, according to the practice of Chen et al. (2015) [54], this article used "family annual net income (took the natural logarithm)" as a substitute variable for the sustainable livelihood ability, and used "human relationship expenses (took the natural logarithm)" as a substitute variable for social capital. As shown in Table 6, the estimation results of the core variables were basically the same in size, significance, and influence direction as those in Table 5. It indicates that the positive impact of social capital on sustainable livelihood ability has a certain degree of stability. Secondly, the heterogeneity of households' social capital is very obvious. In order to ensure the reliability of the conclusion, we not only used a multivariate to measure, but also applied the social capital index to verify. Thirdly, the research used the quantile regression model for verification on the basis of the benchmark regression of "OLS + robust standard error", and the results are basically consistent. In summary, the conclusions of this study are reliable and robust.

**Table 6.** Robustness test.

| Variable | P = 0.10 | P = 0.5 | P = 0.9 |
| --- | --- | --- | --- |
| *S-index* | 0.041 * | 0.026 ** | 0.017 |
| | (0.024) | (0.011) | (0.017) |
| Control variables | yes | yes | yes |
| Pseudo $R^2$ | 0.0129 | 0.0386 | 0.0405 |
| Sample size | | 371 | |

Note. Robust standard errors are in parentheses. The "yes" indicates that control variables are controlled. * Significant at $\alpha = 0.10$, ** Significant at $\alpha = 0.05$, *** Significant at $\alpha = 0.01$.

## 5. Conclusions and Discussion

The purpose of this study was to explore the influence of out-of-poverty households' social capital on their sustainable livelihood ability. This paper used the micro survey data of 371 out-of-poverty households in Jiangxi Province in 2020. We applied the entropy method to calculate the weight of various indicators of social capital, and used "OLS + robust standard error" regression models and quantile regression models for empirical research. The main conclusions are as follows.

Firstly, from the descriptive statistics, the average stock of social capital of out-of-poverty households is generally low. Specifically, the average social capital index of them was 0.33, and the gap was large. From the perspective of social network, the social network owned by households was quite heterogeneous, and few people could help them find

non-agricultural jobs. From the perspective of social participation, about 49% of the out-of-poverty households participated in cooperatives and other organizations. As for social trust, the average social trust of households was 2.96. The level of trust in villagers is average.

Secondly, improving the level of social capital of out-of-poverty households can significantly enhance their sustainable livelihood ability. Specifically, first, social network had a significant positive impact on sustainable livelihood ability. By providing employment opportunities and other social network reciprocal methods, the income of out-of-poverty households can be increased to promote sustainable livelihoods. Second, social participation could significantly promote the improvement of sustainable livelihoods. Out-of-poverty households can support the development of their livelihoods in terms of technology, capacity, and funds through economic organizations such as farmers' cooperatives. Third, it was found that social trust had no significant impact on sustainable livelihoods. The possible reason is that with changes in the social structure of rural areas, such as changes in the relationship between farmers and land, the migration of rural labor, and the development of information and communications, the characteristics of the social capital of rural residents have shown a tendency of differentiation and heterogeneity. Social trust has an obvious tendency to rationalize, which may led to a decline in social trust within the village. Therefore, the influence of social trust on the sustainable livelihood ability of out-of-poverty households is not significant.

Thirdly, social capital has the greatest effect on out-of-poverty households with a low sustainable livelihood ability, which is consistent with the research conclusions of Liu et al. (2016) [58]. Specifically, social capital index, social network, social participation, and social trust all showed significant positive effects on households with a low sustainable livelihood ability. This indicates that social capital plays an important role in preventing marginal households from returning to poverty. This also proves from another aspect that "social capital is the capital of the poor". Therefore, the cultivation of social capital can help narrow the income gap of households. In addition, by observing the coefficients of social capital in each quantile, it was found that its marginal influence basically showed a decreasing trend. This is in line with the law of diminishing returns to scale. At the same time, this also means that with the improvement of households' sustainable livelihood ability. It will be more and more difficult to achieve sustainable development solely through the accumulation of social capital. In other words, social capital needs to work together with other livelihood capitals.

The contributions of this study are mainly reflected in the following three aspects.

First of all, this paper divided social capital into the following three dimensions: social network, social participation, and social trust. Through a multi-dimensional measurement of social capital indicators, the characteristics of social capital of out-of-poverty households were studied more comprehensively while reducing endogenous problems. We found that the social network and social participation of these households had a significant positive impact on their sustainable livelihood ability, while the impact of social trust was not significant. This may be because for out-of-poverty households, social trust cannot be transformed into economic capacity. This is consistent with the research of Wang et al. (2021) [17]. On this basis, we also used quantile regression to obtain the distribution characteristics of social capital for different levels of sustainable livelihood ability. This can find more detailed rules of the influence of social capital than just using "OLS + robust standard error". This study found that social capital contributed the most to households with low sustainable livelihoods. The above findings provide important empirical evidence for the sustainable livelihood of out-of-poverty households in the context of the organic connection between poverty alleviation and rural revitalization.

Secondly, compared with the research objects (general households) of other studies, the research objects of this paper are out-of-poverty households. They are more targeted and difficult to obtain. The sustainability of the livelihoods of this group is related to the consolidation of China's poverty alleviation achievements and the overall revitalization of rural areas.

Thirdly, unlike most previous studies, we consider "the proportion of income other than transfer income to the total net income of the household in 2019" as an index to measure sustainable livelihood ability. The dependent variables in this article are more comprehensive. As for a household with sustainable livelihood ability, wage income should be the main source of income, and operating income and property income should be the potential for sustainable income increase. However, the transfer income belongs to policy income. The lower the proportion of it, the stronger the endogenous development motivation of out-of-poverty households is, and the higher the stability and sustainability of the income is.

Based on the above research conclusions, this paper puts forward the following policy recommendations. To a certain extent, social capital has the attribute of public goods. The government should pay attention to the role of social capital as public goods in promoting the sustainable livelihood of out-of-poverty households in policy and financial support. In addition, the government should actively promote the establishment of social capital as an informal system, and actively promote the accumulation of social capital at the village level and family level of out-of-poverty households. For out-of-poverty households with different income levels, the government should focus on a certain dimension of social capital to cultivate, so as to achieve sustainable livelihoods for them.

Specifically, in terms of social network, rural roads and information construction should be strengthened. They shorten the distance between farmers and the outside world, so that farmers can communicate with society more conveniently. It is necessary to strengthen the internal communication and contact among peasant households. The village committee can build communication platforms among different groups to enable the households out of poverty to fully communicate information and expand their social networks. The government should strengthen the connection between households out of poverty and external entities, such as rural cooperatives, agricultural technology extension stations, and rural supermarkets, to provide support for employment and income.

In terms of social trust, the government can organize a mutual help group between out-of-poverty households and ordinary households. It can promote communication, learning, and mutual assistance among households, thus enhancing the degree of trust. The government should establish and improve interest coordination mechanisms, supervision and management mechanisms, etc., strengthen information transparency, and enhance the trust of households.

In terms of social participation, the government should actively organize households to participate in collective activities. In the process of participating in activities, households will strengthen contact, increase communication, and enhance feelings so as to promote the establishment of social network and trust. The government should mobilize households out of poverty to participate in the management and decision-making of public affairs. This can enhance their ability to participate in self-government and enhance their participation in a cooperative economy.

The livelihood capital of households is a connected organic whole. There is a strong correlation between various types of livelihood capital, and they can be transformed into each other under certain conditions. If the out-of-poverty households want to obtain sustainable livelihood results, they must rationally allocate the five livelihood capitals and give play to their common role. The government can measure the "short-board" livelihood capital of the out-of-poverty households, and carry out governance around the weak links of the households' livelihoods. Therefore, in the process of rural revitalization, according to the specific conditions of households, a certain kind of livelihood capital (social capital) can be used as a starting point to drive the increase in other livelihood capitals.

Although this research helps to deepen the understanding of the relationship between social capital and the sustainable livelihood ability of out-of-poverty households, it still has some shortcomings that need further study. First, the research area of this paper is Nanchang City, Jiangxi Province. Although the sample area includes both mountainous areas and plain areas, the research results of this paper may not be applicable to other

regions. Due to the differences in the economic development level and institutional construction, the effects of social capital dimensions on sustainable livelihood ability may be different in different regions. Secondly, livelihood sustainability is a dynamic process. If dynamic panel data between social capital and sustainable livelihood ability can be obtained, the sustainable characteristics of household livelihood will be discussed more fully.

**Author Contributions:** Conceptualization and writing—review and editing, F.X. (Fangting Xie); project administration, resources, and validation, S.Z.; methodology, formal analysis, and writing—original draft preparation, F.X. (Feixue Xiong); data curation and visualization, H.X.; supervision, X.K. All authors engaged in collecting the data. All authors have read and agreed to the published version of the manuscript.

**Funding:** This study was funded by the National Natural Science Foundation of China (Grant No. 42161053, 71840013 and 41701622), and Jiangxi Province Selenium-rich Agriculture Special Project 2021 (JXFXZD-2021-02).

**Institutional Review Board Statement:** Not applicable.

**Informed Consent Statement:** Not applicable.

**Data Availability Statement:** Not applicable.

**Acknowledgments:** We gratefully acknowledge the investigated households for their time and sincere cooperation and the cadres of 49 villages of Nanchang, Xinjian, Wanli, Jinxian, and Anyi Counties (Districts) for their selfless help in the survey process.

**Conflicts of Interest:** The authors declare no conflict of interest.

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
