# Peer review of "Does Social Capital Benefit the Improvement of Rural Households’ Sustainable Livelihood Ability? Based on the Survey Data of Jiangxi Province, China"

_sustainability, doi:10.3390/su131910995_

Round 1
Reviewer 1 Report
The subject of the article is interesting and fits in the important issue of sustainable development, including poverty alleviation. The aim of the study was to investigate the impact of the social capital of households in poverty on their sustainability. Overall, I noticed that the author (s) put a lot of effort into achieving their research goal. I appreciate their process of developing this research article.
- The summary is correct, it contains all the most important elements.
- Problem formulation: Problem formulation and research gap are well presented.
- The selection, timeliness and review of the content of the literature is sufficient and adequate to the discussed issues.
- The material and research methods have been properly selected and presented sufficiently accurately and clearly.
- Data compiled in tables or presented in other graphic forms constitute the correct documentation of the content of the study.
- The discussion of the results was presented correctly and sufficiently.
- The conclusions were formulated correctly and are confirmed by the results and content of the study.
- Quality of communication: This article has a certain level of readability and is fairly well written.
Author Response
The authors’ Answer: Thank you for your kind comments.

Reviewer 2 Report
This article addresses an important aspect of human development – poverty. It investigates the influence of social capital on sustainable livelihood from a quantitative dimension. Below are my overall impression of the manuscript.
Abstract & Introduction: The abstract broadly represents the content of the manuscript, and the introduction comprehensively gives an orientation of the core background (including definitional concerns) about the subject being investigated. The only missing piece in the introduction is that it abruptly transits to the “theoretical framework” without laying a description of the succeeding sections to allow for overall view of the entire article. What I mean is that the authors could add a small paragraph that describes the rest of the manuscript section by section at the end of the introduction. This will allow for a smooth transition and an understanding of the overall frame of the manuscript earlier.
Theoretical framework: The first sentence, “Based on the Department for International Development (DFID), we proposed an analytical framework for sustainable livelihoods” (lines 142-43) requires a citation as it is difficult to grasp what the authors referred to here. Furthermore, this section is not comprehensive, and the frame argued is more of an analytical framework than a theoretical one. It is important the authors call it what it is – analytical framework, as well boarded the argument further to the poverty discourse. This will allow it to sync with the main frame presented in the introduction.
Methods & analysis: These parts represent the core strength of the manuscript. The key elements of the methods used are represented and identified. They are well explained in details and provides a recognisable standard for replicability.
Discussion & conclusion: This part is well presented. However, I find missing in the discussion, the key policy implications of the results of the analysis in broad terms.
Author Response
Response 1:
Thank you for your kind comments. We agree with you. We have added a small paragraph that describes the rest of the manuscript section at the end of the introduction.
Response 2:
Thank you for your kind comments. We agree with you. We have revised this section as suggested.
Response 3:
Thank you for your kind comments.
Response 4:
Thank you for your kind comments. We have added policy recommendations at the end of the article.

Reviewer 3 Report
This paper is methodologically clear and delves into an interesting topic, the link between social capital and livelihoods in underdeveloped contexts.
In my opinion, it has two major flaws. First, the way to measure the "Trust" component of social capital, as the authors acknowledge, is not appropriate: in terms of social capital, trust should measure the links of confidence among citizens, not towards the State. Also, in a country such as China, respondents are not likely to admit they do not trust the State. So one of the main variables is clearly biased and useless, I wonder if this flaw could be solved.
Second, if the authors have the chance, they should consider the use of a control group, perhaps households that are not out-of-poverty, and if possible, also non-poor households. This could help us see more clearly the role of social capital in out-of-poverty households, otherwise the results could just show a general relationship, not exclusive of this type of households. The quantile analysis goes in this direction, but it is not enough.
In the literature consulted, there is a clear bias towards Chinese sources. In Italy, for example, there is an extensive literature on social capital and rural areas. There are also empirical works about social capital and livelihoods performed in other areas of the world (e.g. Latin America) that could be included in Table 1.
Author Response
Response 1:
Thank you for your kind comment. We agree with you. We have revised this section. We have changed households’ trust in the government to trust in local villagers. We originally set social capital as the trust in the government, in order to explore whether this trust can make the poverty alleviation work more effective, and then make the livelihoods of out-of-poverty households sustainable.
Response 2:
Thank you for your kind comment. We agree with you. Your proposal is very good. But it’s a pity that we do not have data in this regard. We would also like to compare the difference in the role of social capital between out-of-poverty households and non-poor households. In future research, we will try to study this direction in depth. In order to correspond to the research data and content, we changed the title to "Does Social Capital Benefit the Improvement of Out-of-poverty Households’ Sustainable Livelihood Ability? Based on the Survey Data of Jiangxi Province, China".
Response 3:
Thank you for your kind comment. We agree with you. We have expanded the references as suggested.
Round 2
Reviewer 3 Report
English language revision is required, minor spelling mistakes found.
This manuscript is a resubmission of an earlier submission. The following is a list of the peer review reports and author responses from that submission.